# Sarcomeres Morphology and Z-Line Arrangement Disarray Induced by Ventricular Premature Contractions through the Rac2/Cofilin Pathway

**DOI:** 10.3390/ijms222011244

**Published:** 2021-10-18

**Authors:** Yu-Sheng Lin, Tzu-Hao Chang, Wan-Chun Ho, Shun-Fu Chang, Yung-Lung Chen, Shih-Tai Chang, Huang-Chung Chen, Kuo-Li Pan, Mien-Cheng Chen

**Affiliations:** 1Division of Cardiology, Chang Gung Memorial Hospital, Chiayi 61363, Taiwan; dissertlin@gmail.com (Y.-S.L.); erin05201982@gmail.com (W.-C.H.); cst1234567@yahoo.com.tw (S.-T.C.); pankuoli64@gmail.com (K.-L.P.); 2Graduate Institute of Clinical Medical Sciences, College of Medicine, Chang Gung University, Taoyuan 33305, Taiwan; 3Graduate Institute of Biomedical Informatics, Taipei Medical University, Taipei 11031, Taiwan; kevinchang@tmu.edu.tw; 4Department of Medical Research and Development, Chiayi Chang Gung Memorial Hospital, Chiayi 61363, Taiwan; sfchang@cgmh.org.tw; 5Division of Cardiology, Department of Internal Medicine, Kaohsiung Chang Gung Memorial Hospital, Chang Gung University College of Medicine, Kaohsiung 83301, Taiwan; feymanchen@gmail.com (Y.-L.C.); chc3@cgmh.org.tw (H.-C.C.)

**Keywords:** actin cytoskeleton signaling, cardiomyopathy, Rac2/cofilin, sarcomere, ventricular premature contraction

## Abstract

The most common ventricular premature contractions (VPCs) originate from the right ventricular outflow tract (RVOT), but the molecular mechanisms of altered cytoskeletons of VPC-induced cardiomyopathy remain unexplored. We created a RVOT bigeminy VPC pig model (*n* = 6 in each group). Echocardiography was performed. The histopathological alternations in the LV myocardium were analyzed, and next generation sequencing (NGS) and functional enrichment analyses were employed to identify the differentially expressed genes (DEGs) responsible for the histopathological alternations. Finally, a cell silencing model was used to confirm the key regulatory gene and pathway. VPC pigs had increased LV diameters in the 6-month follow-up period. A histological study showed more actin cytoskeleton disorganization and actin accumulation over intercalated disc, Z-line arrangement disarray, increased β-catenin expression, and cardiomyocyte enlargement in the LV myocardium of the VPC pigs compared to the control pigs. The NGS study showed actin cytoskeleton signaling, RhoGDI signaling, and signaling by Rho Family GTPases and ILK Signaling presented z-scores with same activation states. The expressions of Rac family small GTPase 2 (Rac2), the p-cofilin/cofilin ratio, and the F-actin/G-actin ratio were downregulated in the VPC group compared to the control group. Moreover, the intensity and number of actin filaments per cardiomyocyte were significantly decreased by Rac2 siRNA in the cell silencing model. Therefore, the Rac2/cofilin pathway was found to play a crucial role in the sarcomere morphology and Z-line arrangement disarray induced by RVOT bigeminy VPCs.

## 1. Introduction

Ventricular premature contractions (VPCs), presenting as solitary ectopic beats, couplets, or even short runs, are commonly seen in daily clinical practice [1]. VPCs are usually thought to be relatively benign in the absence of structural heart disease, but some observation studies have found that the frequency of VPCs was correlated with sudden cardiac death and heart failure [2]. In patients without structure heart disease, 40% of the idiopathic VPCs were found to be originated from the right ventricular outflow tract (RVOT) [3]. A reversible cause of nonischemic cardiomyopathy due to idiopathic ventricular arrhythmias in the absence of a detectable underlying heart disease has been referred to as VPC-induced cardiomyopathy [4,5]. The left ventricular ejection fraction (LVEF) of VPC-induced cardiomyopathy is less than 50%, and an improvement of LVEF > 15% can be observed following the effective treatment of index ventricular arrhythmias. Although several risks contribute to VPC-induced cardiomyopathy clinically [2,6,7], a VPC burden of more than 20% was reported to be an important cutoff value for VPC-induced cardiomyopathy that deserved effective treatment, either by antiarrhythmics or catheter ablation [6].

Actin cytoskeletons, composed of sarcomeric and non-sarcomeric cytoskeletons, play a crucial role for contractile function [8]. Several studies have confirmed that cardiac systolic/diastolic function was impaired if altering the actin cytoskeleton [9,10]. In terms of the mechanisms of VPC-induced cardiomyopathy, the postulated molecular mechanisms of VPC-induced cardiomyopathy were usually derived from rapid ventricular pacing animal models and included adverse cellular, cytoskeletal, neurohormonal, and proapoptotic changes, ventricular dyssynchrony, abnormal Ca^2+^ handling, etc. [11,12]. However, the molecular mechanisms of altered actin cytoskeleton remain unexplored. Accordingly, we created an RVOT bigeminy VPC model and used next generation sequencing (NGS) and functional enrichment analysis to systemically explore the molecular regulatory mechanisms and functional biological pathways related to the actin cytoskeleton of VPC-induced cardiomyopathy.

## 2. Results

### 2.1. Study Animals

This study enrolled six male pigs in the RVOT-VPC group, with an average age of 8.3 ± 0.99 months, and six male pigs in the sham control group, with an average age of 8.3 ± 0.98 months. During the procedure for the RVOT VPC model creation, the mean body weight was 31.1 ± 1.67 kg in the sham control group and 31.3 ± 1.53 kg in the RVOT-VPC group (*p* = 0.573). All pigs were sacrificed at 6-month follow-up after surgery. On the day of sacrifice, there was no significant difference in the body weight between the two groups (mean body weight was 38.1 ± 3.04 kg in the sham control and 39.1 ± 3.65 kg in the RVOT-VPC group; *p* = 0.417).

### 2.2. Hemodynamics and Echocardiographic Parameters

There were no significant differences in the hemodynamics at the time of operation and before sacrifice between RVOT-VPC group and control group. (Appendix A) However, the RVOT-VPC group had bigger heart weight compared to control group (RVOT-VPC group versus sham control group: 215 ± 4.92 g vs. 178 ± 4.27 g, *p* < 0.001) (Figure 1). Echocardiograms were obtained twice, the first within 1 week before bigeminy VPC creation and the second within 1 week before pigs being sacrificed. There were no significant differences in the LVEDD, LVEF, and LV mass index between the RVOT-VPC group and control group before operation. Of note, at 6-month follow-up, the RVOT-VPC group had significantly larger LVEDD (RVOT-VPC group versus sham control group: 44.47 ± 0.78 vs. 37.78 ± 1.36 mm, *p* = 0.0101) and a higher LV mass index (RVOT-VPC group versus sham control group: 119.6± 11.0 vs. 93.9 ± 6.21 g/m^2^, *p* = 0.0411) than the control group (Figure 1). The RVOT-VPC group had a non-significantly lower LVEF (RVOT-VPC group versus sham control: 57.27± 1.14 vs. 60.74 ± 1.29%) than the control group at 6-month follow-up.

### 2.3. Cardiomyocyte Structure Markers, Cardiomyocyte Morphology, Sarcomere Morphology, and Shear Angle at the Z-Line in the Pig LV Myocardium

Serum cardiac troponin I, a sarcomere biomarker, was measured in the study pigs. The level of serum cardiac troponin I was elevated in the RVOT-VPC group (*n* = 6) (Baseline versus 6-month follow-up: 89.22 ± 28.89 vs. 177.3 ± 71.78 pg/mL, *p* = 0.309) but not in sham control group (*n* = 6) (Figure 2A).

In terms of the morphology of the cardiomyocytes, the cardiomyocytes of the RVOT-VPC group were significantly larger than those of the sham control group at both the LV septum (RVOT-VPC group versus sham control group: 576.0 ± 77.84 vs. 336.6 ± 11.81 μm^2^, *p* = 0.0043) and LV free wall (RVOT-VPC group versus sham control group: 506.0 ± 34.23 vs. 371.6 ± 44.47 μm^2^, *p* = 0.0411) (Figure 2B). In detailed architecture, the cardiomyocyte is composed of myofibrils, each of which contains myofilaments. Myofilaments contain myosin and actin, which anchor to the Z-line. In addition, the sarcomere lies between two Z-lines. Therefore, we further assessed the morphology of sarcomere, and we found that the sarcomere morphology and Z-line arrangement became disarrayed and spindle-shape actin accumulated in the sarcomeres (Figure 3A). The distance between two Z-lines (sarcomere length) of the VPC pigs were larger than the sham control pigs (RVOT-VPC group versus sham control group at septum: 1.80 ± 0.02 vs. 1.57 ± 0.00 μm, *p* < 0.001; RVOT-VPC group versus sham control group at free wall: 1.88 ± 0.02 vs. 1.56 ± 0.00 μm, *p* < 0.001) (Figure 3B,C). Moreover, the shear angles at the Z-line (an indicator of sarcomere with altered cytoskeleton and reduced Z-direction contraction) were increased in the VPC pigs compared to the sham control pigs (RVOT-VPC group versus sham control group at septum: 12.23 ± 0.45 vs. 3.73 ± 0.21 degree, *p* < 0.001; RVOT-VPC group versus sham control group at free wall: 13.68 ± 0.46 vs. 4.09 ± 0.23 degree, *p* < 0.001) (Figure 3B,D), indicating reduced Z-direction contraction and altered torsional mechanics that might result in inefficient contractile function, hypertrophy, and remodeling of the cardiomyocytes of VPC pigs [10]. Ehler et al. demonstrated that the components in intercalated discs can be altered in failing hearts, which includes the altered expression of β-catenin at the adherens junctions and neighboring anchoring structures, from a dilated cardiomyopathy mouse model [13,14]. β-catenin is involved in the regulation and coordination of cell–cell adhesion, and alterations in the localization and expression levels of β-catenin have been associated with dilated cardiomyopathy. Our immunohistological and immunoblotting studies also showed that the spindle-shape actin accumulation (Figure 4A,B) was located at the intercalated discs with β-catenin, and the expression of β-catenin increased in the RVOT-VPC group at both LV septum (Figure 4A,C) and free wall (Figure 4B,D) (β-catenin/GAPDH: RVOT-VPC group versus sham control group at LV septum: 0.96 ± 0.059 vs. 0.64 ± 0.057, *p* = 0.009; RVOT-VPC group versus sham control group at LV free wall: 0.84 ± 0.063 vs. 0.60 ± 0.057, *p* = 0.026). Therefore, we used NGS and functional enrichment analysis of the pig LV myocardium to study the molecular pathway, genes, and protein changes related to the organization of the actin cytoskeleton induced by RVOT VPCs.

### 2.4. Next Generation Sequencing and Functional Enrichment Analysis of the Pig LV Myocardium

The midmyocardial layer of the septum and free wall of the LV at the level of the papillary muscle were dissected after sacrifice at 6-month follow-up in the RVOT-VPC and sham control groups. To identify the profile for the differentially expressed genes (DEGs) in the LV of the VPC pigs, a total of three different LV septal and free wall tissues derived from three VPC pigs were compared with three different LV septal and free wall tissues derived from three sham control pigs. A total of 634 genes were identified to have more than a 1.5-fold change in the LV septal tissues between the VPC pigs and the sham control pigs, i.e., a total of 269 genes were differentially upregulated by the altered fold change > 1.5 (Log_2_FC (RVOT-VPC/sham control) > 0.58) (Appendix A), and 365 were differentially downregulated by the altered fold change < 0.66 (Log_2_FC (RVOT-VPC/sham control) < −0.58) (Appendix A). A total of 557 genes were identified to have more than a 1.5-fold change in the LV free wall tissues between the VPC pigs and the sham control pigs, i.e., a total of 330 genes were differentially upregulated by the altered fold change > 1.5 (Log_2_FC (RVOT-VPC/sham control) > 0.58) (Appendix A), and 227 genes were differentially downregulated by altered fold change < 0.66 (Log_2_FC (RVOT-VPC/sham control) < −0.58) (Appendix A). The heat map graphs are depicted in Appendix A.

To elucidate the molecular mechanisms of the RVOT bigeminy VPC on LV structural remodeling, Ingenuity^®®^ Pathway Analysis (IPA, QIAGEN, Redwood City, CA, USA) software [15] was used for the functional enrichment analysis of the DEGs. There were only six canonical pathways that were identified both in the LV septal and LV free wall tissues, which were actin cytoskeleton signaling, RhoGDI signaling, signaling by Rho Family GTPases, integrin-linked kinase (ILK) signaling, liver X receptor/retinoid X receptor (LXR/RXR) activation, and cardiac hypertrophy signaling (Appendix A and Appendix A). Furthermore, we employed an activation z-score analysis to measure the activation states (activation or inhibition) of the pathways affected by the DEGs, and only four canonical pathways presented z-scores with same activation states in both the LV septum and LV free wall, which were actin cytoskeleton signaling, RhoGDI signaling, signaling by Rho Family GTPases, and ILK Signaling (Table 1 and Appendix A). However, LXR/RXR activation and cardiac hypertrophy signaling had opposite activation states in the LV septum and LV free wall (Table 1 and Appendix A).

RhoGDI signaling regulates Rho Family GTPases signaling [16]. Rho family GTPases have been shown to contribute to the organization of the actin cytoskeleton and of the associated sites of cell adhesion to the extracellular matrix [17]. ILK has been shown to serve as a critical upstream regulator of Rho GTPases [18]. Therefore, the four canonical pathways were all involved in the organization of the actin cytoskeleton, which are consistent with the histopathological changes in the LV myocardium of RVOT-VPC pigs.

### 2.5. mRNA and Protein Expression of the Actin Cytoskeleton Signaling in the Pig LV Myocardium

According to the histopathological changes and functional enrichment analysis of DEGs in the myocardium of the LV septum and LV free wall, we specifically studied the DEGs in the actin cytoskeleton signaling pathway. There were five DEGs in both the LV septum and the LV free wall, which were Rac family small GTPase 2 (Rac2), actin, gamma 2, smooth muscle, enteric (ACTG2), cytoplasmic FMR1 interacting protein 2 (CYFIP2), fibroblast growth factor 16 (FGF16), and integrin subunit alpha 4 (ITGA4). The expressions of Rac2, ACTG2, and ITGA4 were downregulated, while the expressions of CYFIP2 and FGF16 were upregulated in the RVOT-VPC group compared to the sham control group (Figure 5).

Rac2 is a member of the Rac subfamily of the Rho family of GTPases, and Rac1 and Rac2 have both overlapping and distinct roles in actin organization and cell survival [19]. Rac/cofilin plays an important role in actin dynamics and actin polymerization and depolymerization [20]. In addition, cofilin phosphorylation plays an important role in the dynamics of actin polymerization [21]. Therefore, we studied the expressions of Rac2, p-cofilin/cofilin and F-actin/G-actin in both the LV septum and free wall. At 6-month follow-up, the expressions of Rac2 (relative intensity (Rac2/GAPDH): RVOT-VPC group versus sham control group at septum: 0.73 ± 0.02 vs. 0.83 ± 0.03, *p* = 0.0411; RVOT-VPC group versus sham control group at free wall: 0.72 ± 0.01 vs. 0.92 ± 0.06, *p* = 0.048), p-cofilin/cofilin ratio (RVOT-VPC group versus sham control group at septum: 0.54 ± 0.02 vs. 0.69 ± 0.03, *p* = 0.0043; RVOT-VPC group versus sham control group at free wall: 0.78 ± 0.04 vs. 1.21 ± 0.02, *p* = 0.0022), and F-actin/G-actin ratio (RVOT-VPC group versus sham control group at septum: 1.34 ± 0.04 vs. 1.56 ± 0.08, *p* = 0.041; RVOT-VPC group versus sham control group at free wall: 2.10 ± 0.09 vs. 2.63 ± 0.06, *p* = 0.0022) were downregulated in the RVOT-VPC group compared to the sham control group (Figure 6), indicating more actin depolymerization in the LV myocardium of the RVOT VPC pigs that was consistent with the histopathological changes in Figure 3 and Figure 4.

### 2.6. Validation of the Role of Rac2 in Actin Depolymerization in the Rat Ventricular Cell Model

Rat ventricular cardiomyocytes (SV−40 strain) received siRNA to suppress Rac2 expression. The Rac2 protein expression was significantly decreased in the siRac2 group compared to the control group (siCTL group) (relative intensity (Rac2/GAPDH): siRac2 versus siCTL: 0.23 ± 0.04 vs. 0.34 ± 0.03, *p* = 0.049) (Figure 7A). Similar to the results of the RVOT bigeminy VPC pig model, the ratio of p-cofilin/cofilin (siRac2 versus siCTL: 0.04 ± 0.01 vs. 0.093 ± 0.02, *p* = 0.024) and the mean fluorescence intensity of F-actin (siRac2 versus siCTL: 25,048 ± 263.3 vs. 27,851 ± 349.8, *p* < 0.001) were lower in the siRac2 group compared to the siCTL group (Figure 7B,C). As a result of the siRNA to suppress Rac2 expression, actin disarray developed in the siRac2 group compared to the siCTL group (Figure 8A). Moreover, the intensity of actin and the number of actin myofilaments per cardiomyocyte were significantly decreased by the Rac2 siRNA (Figure 8B,C). These findings indicated that Rac2/cofilin played a crucial role in the actin dynamics and actin polymerization and depolymerization.

## 3. Discussion

This RVOT bigeminy VPC pig model was designed to assess the mechanisms of RVOT VPC-induced cardiomyopathy. The histopathological study of LV myocardium of the VPC pigs showed that the sarcomere morphology and Z-line arrangement became disarrayed and spindle-shaped actin accumulated at the intercalated discs that was observed in the failing hearts. [14] Additionally, NGS and the functional enrichment analysis of the DEGs of the LV myocardium induced by RVOT bigeminy VPCs identified four canonical pathways with same activation states in both the LV septum and the LV free wall, i.e., actin cytoskeleton signaling, RhoGDI signaling, signaling by Rho Family GTPases and ILK Signaling. All the four canonical pathways were involved in the organization of the actin cytoskeleton. Moreover, studies of gene and protein expression related to the actin cytoskeleton signaling in the pig LV myocardium showed that the expressions of Rac2, the p-cofilin/cofilin ratio, and the F-actin/G-actin ratio were downregulated in the RVOT-VPC group compared to sham control group after 6-month RVOT bigeminy VPCs. Furthermore, a cell study using rat ventricular cardiomyocytes (SV-40 strain) confirmed that Rac2/cofilin played a crucial role in actin dynamics and actin polymerization and depolymerization.

### 3.1. Actin Cytoskeleton Organization and Cardiac Function

Most previous studies of the mechanisms of arrhythmia-related cardiomyopathy mainly investigated pathways related to calcium handling, calcium sensitization and alteration of structural proteins, including adhesion proteins and myosin. However, cardiac systolic dysfunction already existed in these prior studies, indicating late-stage changes [12]. In our study, the calcium signaling pathway (*p* value = 0.569) was not selected by functional enrichment analysis of DEGs by IPA with the selection criteria of DEGs by the fold change (Log_2_FC > 0.58 or Log_2_FC < −0.58) and *p* value less than 0.05. In our study, heart size was increased after 6-month RVOT bigeminy VPCs but LV systolic function was not significantly depressed (Figure 1). Therefore, the findings of our model were representative of an earlier stage of myocardial changes induced by RVOT bigeminy VPCs.

Actin cytoskeletons, composed of sarcomeric and non-sarcomeric cytoskeletons, play a crucial role in contractile function, signal transduction, cell proliferation, and cell growth [8]. Cardiac contraction force is generated by interactions between actin and myosin in the sarcomere, and altered sarcomere with reduced Z-direction contraction and actin cytoskeleton architecture disarray in the ventricular wall may contribute to impairments in the torsional mechanics that underlie heart failure [10]. Several studies have confirmed that cardiac systolic/diastolic function was impaired if altering the actin cytoskeleton [9,10]. One prior study reported that thin filament (F-actin) elongation by the increased expression of Lmod2 resulted in heart failure in a mouse model [22]. In our study, sarcomere lengths of the RVOT-VPC pigs were significantly larger than the sham control pigs. Moreover, the shear angles at the Z-lines were increased in the RVOT-VPC pigs compared to the sham control pigs. Furthermore, the expression of F-actin/G-actin ratio was downregulated in the RVOT-VPC group compared to the sham control group. All these molecular and pathological findings could lead to cardiomyopathy, as shown in previous heart failure animal models of disorganized sarcomeric actin [23].

### 3.2. The Crucial Role of Rac2 in Actin Cytoskeleton Organization

Rac2/cofilin plays an important role in actin dynamics and actin polymerization and depolymerization [20]. In addition, cofilin phosphorylation plays an important role in the dynamics of actin polymerization [21]. In our study, the RVOT bigeminy VPC pig model and the Rac2 silencing cell model revealed that Rac2/cofilin plays a crucial role in actin dynamics and actin polymerization and depolymerization.

### 3.3. Limitations

Some limitations existed in our RVOT bigeminy VPC pig model. First, the impact of the small current RV endocardial pacing effect on the pathophysiological changes of LV following 6-month RVOT bigeminy VPCs could not be ignored. However, in order to avoid the influence of the endocardial pacing current on pathophysiological changes, the selected dissected tissues of the LV septum and the LV free wall for study were far from the endocardial pacing site. Second, although our study revealed that Rac2 played a crucial role in actin dynamics and actin polymerization and depolymerization, these findings were not further examined in the Rac2 knock-out pig model. Third, ROCK/LIMK/cofilin pathway, one downstream pathway of the four studied canonical pathways, plays an important role in regulating actin cytoskeleton organization [24]. Although our results showed that Rac2/cofilin played a crucial role in actin cytoskeleton disarray induced by RVOT bigeminy VPC, the role of other aforementioned pathways or mechanisms which possibly involve channelopathy, beta-adrenergic receptors, and so on, in VPC-induced cardiomyopathy was not well elucidated and merits further study.

## 4. Method

### 4.1. Study Animals and the Creation of a RVOT Bigeminy VPCs Model

This study enrolled 12 male Lanyu miniature pigs, which were divided into two groups: a RVOT-VPC group (*n* = 6) and a sham control group (*n* = 6). The flowchart of the study procedure is shown in Appendix A. All pigs received appropriate care as defined by the Guide for the Care and Use of Laboratory Animals, published by Taiwan’s National Institutes of Health (NIH publication No. 85-23, revised in 1996), and the animal procedures were approved by the Institutional Animal Care and Committee (IACUC) at Chang Gung Memorial Hospital (IACUC Number: 2015112701). All procedures conformed to the guidelines from the NIH Guide for the Care and Use of Laboratory Animals. All procedures and echocardiography measurements were performed under general anesthesia and after food restriction for 12 h and water restriction for 4 h. All pigs (average age of 8–10 months) of the RVOT-VPC group received a surgical procedure for the placement of dual chamber pacemaker at the left paratracheal area and pacing leads at RVOT and RV free wall for ventricular pacing and sensing, respectively, to create bigeminy VPC. (Appendix A).

### 4.2. Creating RVOT Bigeminy VPCs by Pacing Leads and Pacemaker Setting

All pigs received the surgical procedure under general anesthesia with 2.0–4.0% isoflurane through endotracheal intubation and premedication with atropine (1 mg/kg), ketamine (12 mg/kg), and xylazine (1.2 mg/kg) by intramuscular injection. After general anesthesia was performed, an ear vein was punctured for the placement of a catheter for an intravenous infusion with 500 mg of cefamezine, and then the pacemaker and lead placement procedure followed. A pacemaker pocket was created over the left paratracheal area and was approximately 4 cm in length. Pacing leads were inserted into an external jugular vein using the cutdown method. An RV screw-in bipolar lead (St Jude) was first inserted into the RV under fluoroscopic guidance and was fixed over the RVOT septum (Appendix A) after two criteria were met—one was that the threshold was less than 1 V under the pulse width of 0.4 msec and the R wave amplitude was higher than 4 mV; the other was that the electric axes of VPC showed positive QRS complexes in leads II, III, and aVF. Another screw-in bipolar lead (St Jude), which served as the sensing lead, was then inserted into the right ventricle and fixed over the free wall of RV under fluoroscopic guidance after confirming that the R wave amplitude was higher than 4 mV. After the extravenous portions of the leads were fixed over the paratracheal muscle and the generator was connected to the leads, 1 g of vancomycin powder was scattered over the pacemaker pocket. A two-layer method was employed to close the surgical wound. The generator was set to the VDD (ventricular pacing and dual sensing) mode and ventricular pacing was set at 425 msec coupling interval after sensing intrinsic ventricular signal to create bigeminy VPC; the output setting was two times the threshold, and the sensitivity setting was 0.5 times the R wave (sensing lead). During the 6-month follow-up period, all generators were interrogated at 1 week, 1 month, 3 months, and before sacrifice to ensure >99% RVOT bigeminy VPC pattern achieved (Appendix A). Pigs of the sham control group received a surgical procedure for the placement of pacing leads at RVOT and RV free wall but without pacemaker placement. The hearts of all pigs were resected for tissue, molecular, and genetic studies through thoracotomy after euthanasia with general anesthesia at the 6-month follow-up.

### 4.3. Hemodynamics and Transthoracic Echocardiograph

Hemodynamics were measured by 5-F pig-tail catheter before placement of leads and sacrifice, which included mean LV pressure and mean ascending aortic pressure measured through the left femoral artery and mean pressure in right atrium, RV, and pulmonary artery through the right external jugular vein. All echocardiographic procedures were performed using a commercially available echocardiography system (Vivid 7; GE-Vingmed, Horten, Norway). Two-dimensional (2D), 2D strain, and M-mode echocardiography was performed in the right parasternal area at a left lateral decubitus position after 10 min when pacemaker was switch-off. The LV end-diastolic diameter (LVEDD), LV end-systolic diameter, and end-diastolic and end-systolic thickness of the LV posterior wall were measured using the M-mode under the right parasternal long-axis view, as recommended by the American Society for Echocardiography [25]. The LV mass index and LVEF were computed according to parameters from M-mode [26].

### 4.4. ELISA for Cardiac Troponin I

Blood samples from all study pigs were obtained before surgery and at sacrifice. The serum was immediately separated and frozen at −80 °C until the assay. Enzyme-linked immunosorbent assay was applied to measure the serum-EDTA concentrations of cardiac troponin I using ELISA kits (Fine Test, Wuhan, China). Determination of troponin I levels was carried out according to the manufacturer’s instruction.

### 4.5. Specimen Storage

The LV free wall and septal tissues at the papillary muscle level of the pigs were obtained after euthanasia under general anesthesia. Some ventricular tissues were immediately frozen in liquid nitrogen at −80 °C for ribonucleic acid (RNA) analyses. Additionally, some ventricular tissues were placed into a tissue Tek^®®^ container, which was then filled with Tissue-Tek^®®^ optimum cutting temperature compound (Sakura Finetek, CA, USA); these samples were frozen in liquid nitrogen for later histochemical analysis, and some were immediately fixed in 3.7% buffered formalin and then embedded in paraffin for histological study.

### 4.6. Confocal Immunofluorescence Study

Pig ventricular tissues and rat ventricular cardiomyocytes (SV-40 strain) were fixed by 4% paraformaldehyde in PBS for 10 min at room temperature, then were washed with PBS for 5 min × 3, treated by 0.1%Triton X-100 in PBS for 10 min, and blocked with 1% BSA in PBS and then were treated with alpha–actinin antibody (1:100, Abcam, Cambridge, MA, USA) and ß-catenin (1:200, BD, San Diego, CA, USA) overnight at 4 °C. Primary antibody was detected with Alexa-Fluor 555 or 647-conjugated antibodies (Abcam, MA, USA). Nuclei were visualized with DAPI (Invitrogen, Carlsbad, CA, USA) and F-actin with Phalloidin-Alexa-Fluor-488-conjugate (Invitrogen, CA, USA). Immunofluorescence-labeled samples were examined by using Leica TCS SP5 II confocal laser-scanning microscopy (Leica, Wetzlar, Germany). All samples were analyzed by IMAGE J (National Institutes of Health).

### 4.7. Image Analysis

Sarcomere length was defined as the distance between two Z-lines, and the distance of two Z-lines were measured according to the distance of neighboring two peaks of immunofluorescent intensity of alpha–actinin using the DDecon of ImageJ plug-in [27]. The measurement was performed in at least 10 independent areas in a confocal microscopic field. The shear angle at the Z-line was defined as the angle between Z-line and the line parallel to the actin in the same sarcomere, i.e., the angle between α-actinin and F-actin, and the values were averaged from the measurements of least 10 independent areas in a confocal microscopic field [10]. The above measurements for Z-line distance and the shear angle at the Z-line of cardiomyocytes were repeated in 100 individual cells per sample.

To quantify individual myocyte size, LV free wall, and septal tissue sections were stained with Alexa Fluor^®®^ 555 conjugate of wheat germ agglutinin (WGA) (Thermo Fisher Scientific Inc., Rockford, IL, USA). Next, 5-μm histological heart sections were cut and stained with WGA (50 μg/mL) to identify membranes for measuring myocyte cross-sectional area. Myocyte area was assessed with ImageJ (National Institutes of Health), and the cell size was calculated from 1000 cells in individual 10 fields of each sample (1 sample per pig).

For actin myofilament quantification in the ventricular cell (SV-strain), we drew three 10-μm long lines at equal separations in three different parts along the long axis of ventricular cells by using ImageJ (spot function). The estimated quantity of actin myofilaments in the ventricular cell was defined as the mean intensity value of the fluorescent signals of F-actin cross the lines and the number of actin myofilaments per cardiomyocyte was defined as the mean number of peaks of fluorescent signals of F-actin cross the lines [8]. The above measurements for actin myofilament quantification in the ventricular cell were obtained from 100 individual cells per sample in each of the 4 repeated experiments.

### 4.8. Western Blot

Protein extracts of LV myocardial tissues and rat ventricular cardiomyocytes (SV-40 strain) were prepared using a CelLytic™ MT Cell Lysis Reagent (Sigma-aldrich, St. Louis, MO, USA) with protease/phosphatase inhibitor. Homogenates were centrifuged at 14,000 rpm for 20 min at 4 °C to yield supernatants. The concentrations of sample proteins were determined using the BCA method (Thermo Fisher Scientific Inc., IL, USA) according to the supplier’s instructions. HeLa whole cell lysate (Santa Cruz, TX, USA) served as positive controls. Protein extracts (30–60 μg) were electrophoresed on 10% to 12% acrylamide SDS-PAGE gel at room temperature for 1 h and electro-transferred onto polyvinylidene difluoride membranes for 1 h on ice. The membranes were blocked at room temperature for 1 h in Tris-buffered saline containing 0.1% Tween-20 and 5% (*w*/*v*) nonfat dry milk or 5% (*w*/*v*) bovine serum albumin. The primary antibodies, including anti-Rac2 (1:500 dilution; Santa Cruz, TX, USA), cofilin (1:10,000 dilution; Abcam, MA, USA), p-cofilin (1:500 dilution; cell signaling, Danvers, MA, USA), and ß-catenin (1:1500, BD, CA, USA) were used to react with the blots at 4 °C overnight in 5% nonfat dry milk. The blots were washed 3 times in Tris-buffered saline containing 0.1% Tween-20 and incubated at room temperature for 1 h with horseradish peroxidase-labeled secondary antibody at dilutions of 1:10,000 in Tris-buffered saline containing 0.1% Tween-20 and 5% nonfat dry milk. Following 3 washes, blots were incubated with Immobilon Western chemiluminescent HRP substrate (Millipore, Burlington, MA, USA). All specific values of evaluated proteins were standardized to anti-GAPDH antibody (1:50,000 dilution; Millipore, Burlington, MA, USA).

### 4.9. RNA Isolation

RNAs were extracted from the myocardial tissue using a RiboPure kit (Ambion, Grand Island, NY, USA) according to the manufacturer’s protocol. The quality of RNA was assessed using an Agilent 2100 Bioanalyzer (Agilent Technologies Inc., Santa Clara, CA, USA), and samples with an optical density ratio 260/280 > 1.8 and RNA integrity number > 7.0 were selected for next generation sequencing (NGS) analysis and real-time reverse transcriptase–polymerase chain reaction (RT-PCR).

### 4.10. Next Generation Sequencing and Enrichment Analysis

Paired-end sequencing was performed using Illumina NovaSeq^TM^ 6000 sequencing system in FASTQ format and sequence quality was assessed using the FastQC. A total of 1315 million paired-end sequence reads were obtained from the 12 samples (LV free wall and septal tissues of 6 pigs, i.e., 3 random pigs from each group) with a median reads length of 139 bp and a median of 55.8 million reads per sample. The QIAGEN CLC Genomics Workbench was used for RNA-Seq and statistical analysis including quality control, reference alignment, gene expression profiling, and differentially expressed gene (DEG) analysis. Sus scrofa reference genome and the annotation files from Ensembl genome database were obtained though CLC bio download module. The reads were trimmed with Phred quality score (Q score) more than 20 and filtered out if the length of each read were shorter than 100 bps. Qualified sequence reads were aligned to Sus scrofa reference genome, and the TPM (transcript per million) of each gene was calculated. Statistical analysis module was applied to identify DEGs between comparison subgroups, and DEGs were selected with a fold change (FC) of >1.5 (Log_2_FC > 0.58) or < 0.66 (Log_2_FC < −0.58) and *p* value less than 0.05. QIAGEN’s Ingenuity^®®^ Pathway Analysis (IPA^®®^, QIAGEN, Redwood City, CA, USA) software [15] was used for functional enrichment analysis of the DEGs. In order to obtain more functional information, the DEGs belonged to orthologous genes between Sus scrofa and Homo sapiens were particularly collected for enrichment analysis. Z-score was applied to measure predicted activation states (<0: inhibition, >0: activation) of the pathways affected by DEGs.

### 4.11. Quantitative Determination of RNAs by Real-Time RT-PCR

RNAs were extracted from the pig LV free wall and septal myocardial tissues and rat ventricular cardiomyocytes (SV-40 strain, ABM Inc., Richmond, BC, Canada) using a RiboPure kit (Ambion, Grand Island, NY, USA) according to the manufacturer’s protocol. First-strand cDNAs were synthesized with reverse transcriptase and oligo (dT) primers. Real-time quantitative PCR was performed on the ABI Prism 7500 FAST sequence detection system (Applied Biosystems, Foster City, CA, USA), using SYBR Green PCR Master Mix (Qiagen, Redwood City, CA, USA). The results were normalized against GAPDH gene expression (endogenous control). We selected 5 genes that were differentially upregulated or downregulated in the RVOT-VPC group compared to sham control group according to NGS analysis results.

### 4.12. G/F Actin Assay

We assayed LV free wall, LV septal tissues, and rat ventricular cardiomyocytes (SV-40 strain) with the G-actin/F-actin in vivo assay kit (Cytoskeleton, Denver, CO, USA). Myocardial tissues and cells were lysed in LAS2 buffer (pH 6.9). After centrifuging the lysate to a pellet of unbroken cells, supernatants were pipetted into clear ultracentrifuge tubes and were centrifuged at 100,000× *g*. This step separated the pellet F-actin (filamentous) and left the G-actin (globular) in supernatant. We analyzed the supernatant (G-actin) and pellet (F-actin) fraction for actin content by immunoblotting with anti-actin antibody.

### 4.13. Cell Culture

Rat ventricular cardiomyocytes (SV-40 strain) were cultured in Prigrow III Medium (ABM Inc., Canada). Culture medium was supplemented with 15% (*vol*/*vol*) fetal bovine serum and 1% penicillin/streptomycin. Cells were incubated at 37 °C in a humidified atmosphere of 5% CO_2_ and 95% air.

### 4.14. Small Interfering RNA (siRNA)-Mediated Knockdown in Cell Model

To suppress Rac2 expression, the silencer selected siRNAs against Rac2 (siRac2: 5′- GUGAAGUACUUGGAAUGUUtt-3′ and 5′-CAGACAGACGUGUUCCUCAtt-3′) was purchased from Ambion (Austin, TX, USA). Rat ventricular cardiomyocytes (SV-40 strain) were concomitantly transfected with both siRac2 using Lipofectamine RNAiMAX Transfection Reagent (Invitrogen, CA, USA) according to the manufacturer’s instructions, and experiments were performed 24 h later.

### 4.15. Measurement of F-Actin in Rat Ventricular Cardiomyocytes by Flow Cytometry

Phalloidin-Alexa-Fluor-488 (Invitrogen, CA, USA) was used to stain F-actin. Rat ventricular cardiomyocytes (SV-40 strain) were plated in 6-well plates and allowed to attach for 24 h. After being transfected for 24 h, the cells were harvested through trypsinization, washed in PBS, fixed by 4% paraformaldehyde in PBS for 10 min at room temperature, and then were washed with PBS for 5 min × 3, treated by 0.1%Triton X-100 in PBS for 10 min, and then suspended in Phalloidin-Alexa-Fluor-488-conjugate. After incubation for 20 min at room temperature, the cells were washed 3 times and suspended in PBS. To determine the intracellular F-actin content using flow cytometry, 10,000 cells per sample were analyzed using the BD FACSCanto II (BD Biosciences, East Rutherford, NJ, USA).

### 4.16. Statistical Analysis

Data are presented as the mean ± standard deviation or standard error of the mean. Continuous variables were analyzed using the Mann–Whitney U test. Statistical analysis was performed using commercial statistical software (SPSS for Windows, version 22; SPSS Inc., Chicago, IL, USA). A *p* value (two-tailed) of <0.05 was considered statistically significant.

## 5. Conclusions

The sarcomere morphology and Z-line arrangement became disarray in the LV myocardium of pigs by 6-month persistent RVOT bigeminy VPCs. The Rac2/cofilin pathway was found to play a crucial role in the sarcomere morphology and Z-line arrangement disarray induced by the RVOT bigeminy VPC, which may underlie VPC-induced cardiomyopathy. Further experimental studies such as knock-out animal models and studies with a larger number of animals are warranted to validate our findings.

## Figures and Tables

**Figure 1 ijms-22-11244-f001:**
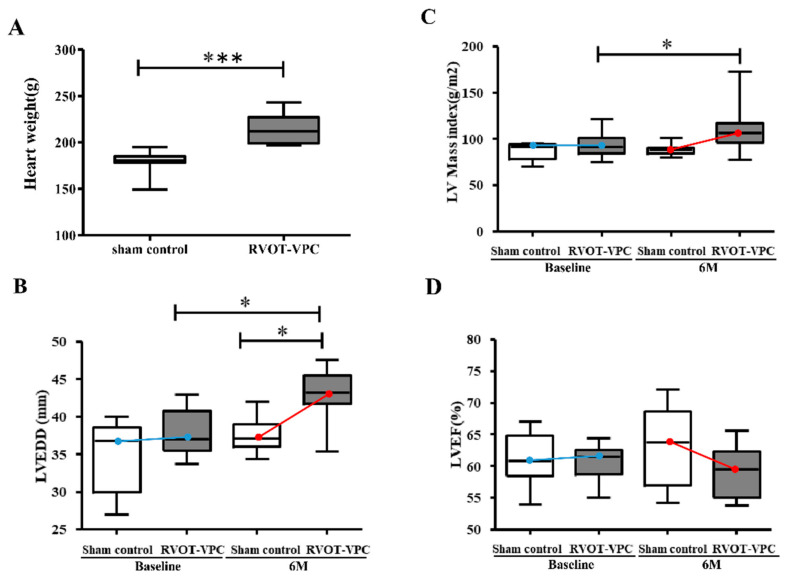
Heart weight, left ventricular mass, left ventricular size, and left ventricular ejection fraction between the RVOT-VPC group and the sham control group at baseline and 6-month follow-up. Heart weight of the RVOT-VPC group was bigger than the sham control group at the 6-month follow-up (*p* < 0.001) (**A**). The left ventricular end-diastolic diameter (LVEDD) of the RVOT-PVC group at the 6-month follow-up was significantly larger than that of the RVOT-PVC group at the baseline (*p* = 0.0022) and that of the sham control group at the 6-month follow-up (*p* = 0.0101) (**B**). The LV mass index of the RVOT-VPC group was significantly higher than that of the sham control group at the 6-month follow-up (*p* = 0.041) (**C**). There was a trend toward lower LV ejection fraction (LVEF) in the RVOT-PVC group compared to the sham control group at the 6-month follow-up (**D**). *: *p* < 0.05; ***: *p* < 0.001. Blue line to express the trend of difference between Sham control and RVOT-PVC groups at baseline; Red line to express the trend of difference between Sham control and RVOT-PVC groups at the 6-month follow-up.

**Figure 2 ijms-22-11244-f002:**
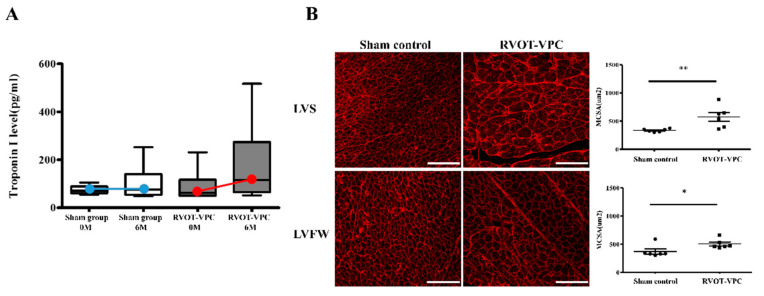
Elevated serum cardiac troponin I level and cardiomyocyte enlargement developed in left ventricular myocardium following 6-month RVOT bigeminy VPCs. (**A**) Before VPC creating, serum cardiac troponin I levels were similar between RVOT-VPC group and sham control group. However, the RVOT-VPC group had higher serum cardiac troponin I level than the sham control group at the 6-month follow-up, although the difference did not reach statistical significance. The changes in cardiac troponin I level in the RVOT VPC group and sham control group were illustrated by the slopes of the red line and blue line, respectively. (**B**) Immunofluorescence study with wheat germ agglutinin stain for the cell membrane of cardiomyocytes showed that the RVOT-VPC group had significantly larger cardiomyocyte size than the sham control group at both the left ventricular septum (LVS) (*p* = 0.0043) and LV free wall (LVFW) (*p* = 0.0041). Round dot: Sham control group; Square dot: RVOT-VPC group. Each datum in this figure represents the average of 1000 cells per sample. MCSA: myocyte cross-sectional area. Scale bar = 75 μm. *: *p* < 0.05; **: *p* < 0.01.

**Figure 3 ijms-22-11244-f003:**
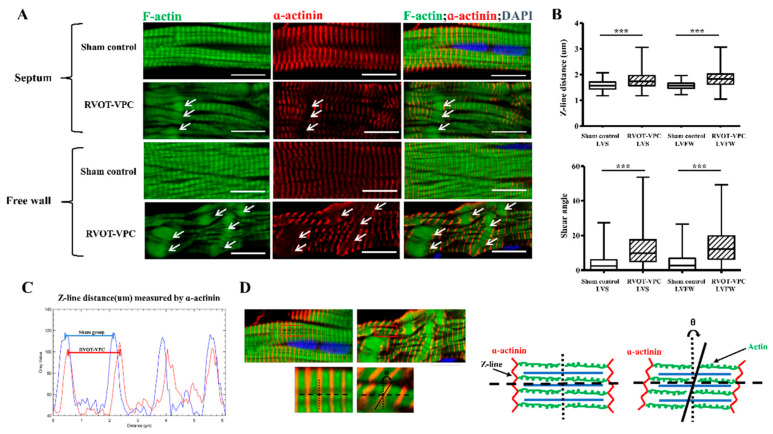
Sarcomere morphology, shear angle at the Z-line and the distance between two Z-lines following 6-month RVOT bigeminy VPCs. (**A**) Immunohistological studies of left ventricular septal and free wall myocardial tissue showed organized sarcomeres (homogenous distribution of F-actin in the sarcomeres, Z-lines in parallel, and equal distance between Z-lines (stained with α-actinin)) in the sham control group but showed that the sarcomere morphology and Z-line arrangement became disarrayed (non-homogenous distribution of F-actin in the sarcomeres and Z-lines not in parallel and different distances between two Z-lines) and spindle-shaped actin accumulations (white arrow) in the RVOT-VPC group. (**B**–**D**). The distance between two Z-lines (sarcomere length) and the shear angle at the Z-line (an indicator of reduced Z-direction contraction) were significantly increased in the cardiomyocytes of RVOT-VPC group compared to sham control group at both locations of left ventricle. The length of sarcomeres was measured as the distance between neighboring Z-lines, i.e., the distance between the neighboring peak immunofluorescent intensities of α-actinin (**C**). The shear angle at the Z-line was defined as the angle between Z-line and the line parallel to the actin in the same sarcomere, i.e., the angle between α-actinin and F-actin (**D**). DAPI staining for nuclei. Scale bar = 10 μm. ***: *p* < 0.001.

**Figure 4 ijms-22-11244-f004:**
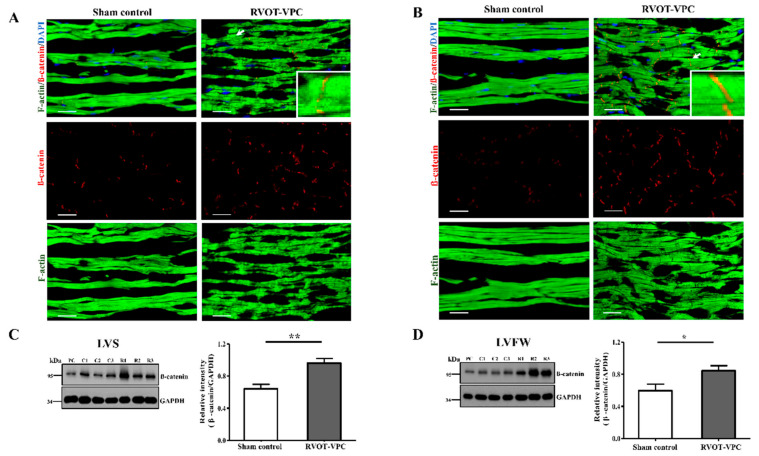
Immunohistological and immunoblotting studies of the expression of β-catenin in the RVOT-VPC group and sham control group at left ventricular septum (LVS **A**,**C**) and left ventricular free wall (LVFW **B**,**D**. The expression of β-catenin at intercalated discs significantly increased in the RVOT-VPC group than in the sham control group at LVS (**A**,**C**) and LVFW (**B**,**D**). In addition, the spindle-shape actin accumulation was located at the intercalated discs with β-catenin (insert) (white arrow at right upper panel in (**A**,**B**). DAPI staining for nuclei. Scale bar = 10 μm. *: *p* < 0.05 and **: *p* < 0.01. PC: positive control.

**Figure 5 ijms-22-11244-f005:**
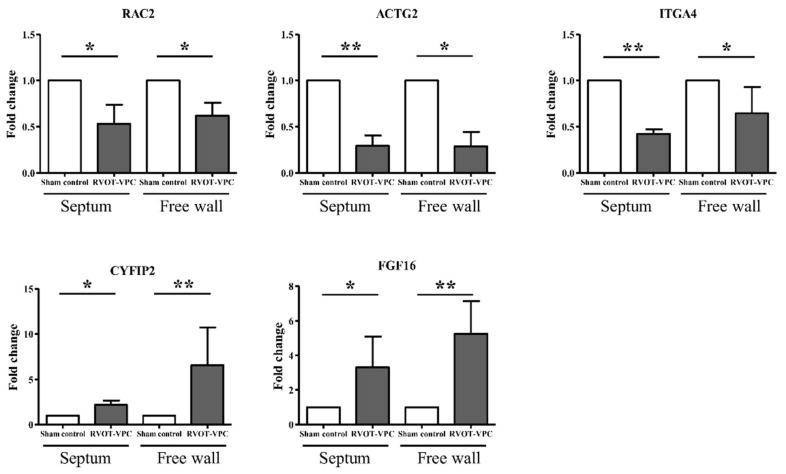
Five differentially expressed genes related to actin cytoskeleton signaling induced by 6-month RVOT bigeminy VPCs in the septum and free wall of left ventricle. Five genes that were differentially upregulated or downregulated in the RVOT-VPC group compared to the sham control group according to next generation sequencing analysis results were selected and studied. The expressions of RAC2, ACTG2, and ITGA4 were downregulated while the expressions of CYFIP2 and FGF16 were upregulated in the RVOT-VPC group compared to sham control group. *: *p* < 0.05 and **: *p* < 0.01.

**Figure 6 ijms-22-11244-f006:**
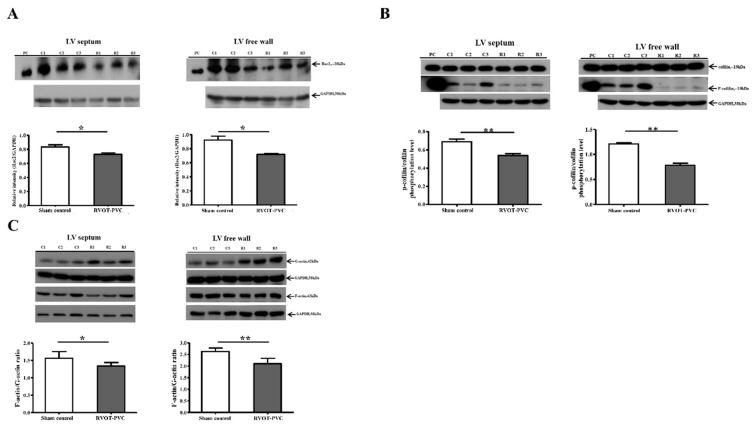
The expression of Rac2, phosphorylation of cofilin and F-actin/G-actin ratio in the LV myocardium following 6-month RVOT bigeminy VPC. Immunoblotting study showed that the expressions of (**A**) Rac family small GTPase 2 (Rac2), (**B**) Ratio of p-cofilin/cofilin, and (**C**) F-actin/G-actin ratio in left ventricular (LV) septum and LV free wall were significantly decreased in the RVOT bigeminy VPC group compared to the sham control group at the 6-month follow-up. *: *p* < 0.05; **: *p* < 0.01. PC: positive control.

**Figure 7 ijms-22-11244-f007:**
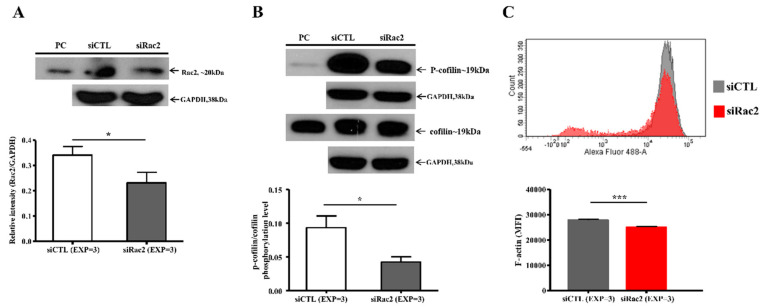
The expression of the Rac2, ratio of p-cofilin/cofilin and F-actin in the rat ventricular cardiomyocytes after silencing Rac2 gene. After silencing Rac2, the expression of Rac2 protein in the rat ventricular cardiomyocytes was significantly decreased compared to the control group (siCTL) without silencing Rac2 by immunoblotting study (**A**). The ratio of p-cofilin/cofilin in the rat ventricular cardiomyocytes was significantly lower in the siRac2 rat ventricular cardiomyocytes than the siCTL group (**B**). Flow cytometric study showed the mean fluorescence intensity (MFI) of F-actin was significantly lower in the siRac2 rat ventricular cardiomyocytes compared to the siCTL group (**C**). *: *p* < 0.05 and ***: *p* < 0.001. PC: positive control.

**Figure 8 ijms-22-11244-f008:**
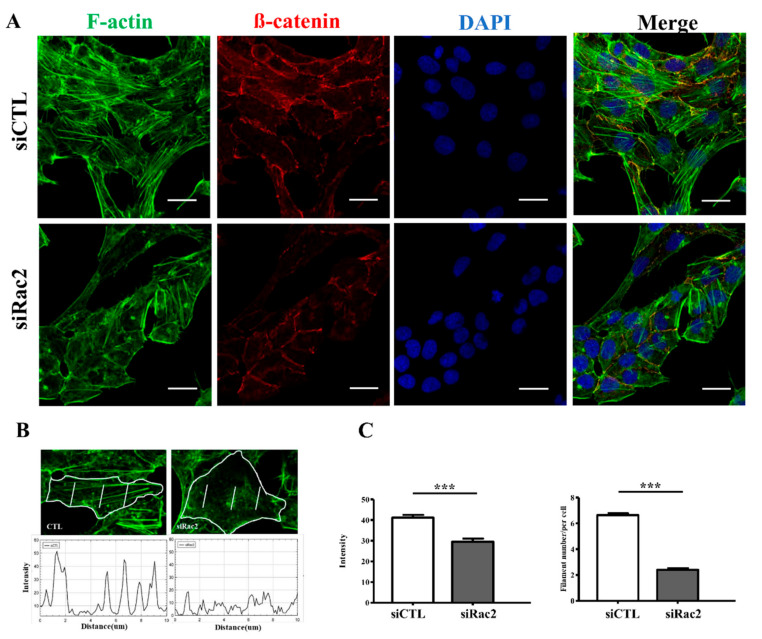
The expression of actin filament in terms of intensity and number of actin filaments in the rat ventricular cardiomyocytes with and without silencing Rac2 gene by confocal immunofluorescence study. (**A**) Silencing Rac2 gene to suppress Rac2 expression significantly changed the actin distribution in the rat ventricular cardiomyocytes. ß-catenin staining for cell membrane; DAPI staining for nuclei. (**B**) Three 10-μm lines were drawn at equal distance at three separate parts of cardiomyocyte and the mean immunofluorescence intensity and number of immunofluorescence peaks were calculated by using ImageJ (spot function) and these values were measured from 100 individual cells per sample in each of the four repeated experiments. (**C**) The intensity of actin (F-actin) and number of actin myofilaments per cardiomyocyte were significantly decreased by Rac2 siRNA. Scale bar = 20 μm. ***: *p* < 0.001.

**Table 1 ijms-22-11244-t001:** Log_2_ fold change values and predictive activity of the differentially expressed genes significantly involved in the actin cytoskeleton signaling pathway.

Tissue	Pathways	Symbol	Entrez Gene Name	Z Score	Log_2_FC Value	Predictive Activity to Pathway (IPA Knowledge Base)
**Septum**	Actin Cytoskeleton Signaling	ACTG2	actin, gamma 2, smooth muscle, enteric	−1.265	−2.112	Activation
		CYFIP2	cytoplasmic FMR1 interacting protein 2		0.855	-
		FGF16	fibroblast growth factor 16		2.58	-
		ITGA4	integrin subunit alpha 4		−1.28	Activation
		RAC2	Rac family small GTPase 2		−0.876	Activation
	Signaling by Rho Family GTPases	ACTG2	actin, gamma 2, smooth muscle, enteric	−1.508	−2.112	Activation
		CDH12	cadherin 12		2.135	Activation
		CDH24	cadherin 24		−1.45	Activation
		GNB3	G protein subunit beta 3		1.457	-
		ITGA4	integrin subunit alpha 4		−1.28	Activation
	RhoGDI Signaling	ACTG2	actin, gamma 2, smooth muscle, enteric	0.333	−2.112	inhibition
		CDH12	cadherin 12		2.135	inhibition
		CDH24	cadherin 24		−1.45	inhibition
		GNB3	G protein subunit beta 3		1.457	-
		ITGA4	integrin subunit alpha 4		−1.28	inhibition
	ILK Signaling	ACTG2	actin, gamma 2, smooth muscle, enteric	−1.414	−2.112	Activation
		FBLIM1	filamin binding LIM protein 1		0.636	-
		MMP9	matrix metallopeptidase 9		−2.105	-
	LXR/RXR Activation	ARG2	arginase 2	2.646	1.931	Activation
		MMP9	matrix metallopeptidase 9		−2.105	inhibition
		S100A8	S100 calcium binding protein A8		−3.753	inhibition
		SERPINF1	serpin family F member 1		1.384	Activation
	Cardiac Hypertrophy Signaling	ADRB2	adrenoceptor beta 2	−0.632	0.804	Activation
		CACNA1S	calcium voltage-gated channel subunit alpha1 S		−4.149	Activation
		GNB3	G protein subunit beta 3		1.457	Activation
**Free wall**	Actin Cytoskeleton Signaling	ACTG2	actin, gamma 2, smooth muscle, enteric	−1.633	−2.595	Activation
		CYFIP2	cytoplasmic FMR1 interacting protein 2		1.046	-
		FGF16	fibroblast growth factor 16		1.397	-
		ITGA4	integrin subunit alpha 4		−0.937	Activation
		RAC2	Rac family small GTPase 2		−0.814	Activation
	Signaling by Rho Family GTPases	ACTG2	actin, gamma 2, smooth muscle, enteric	−0.632	−2.595	Activation
		CDH12	cadherin 12		1.983	Activation
		CDH24	cadherin 24		−1.144	Activation
		GNB3	G protein subunit beta 3		0.991	-
		ITGA4	integrin subunit alpha 4		−0.937	Activation
	RhoGDI Signaling	ACTG2	actin, gamma 2, smooth muscle, enteric	0.707	−2.595	inhibition
		CDH12	cadherin 12		1.983	inhibition
		CDH24	cadherin 24		−1.144	inhibition
		GNB3	G protein subunit beta 3		0.991	-
		ITGA4	integrin subunit alpha 4		−0.937	inhibition
	ILK Signaling	ACTG2	actin, gamma 2, smooth muscle, enteric	−1.134	−2.595	Activation
		FBLIM1	filamin binding LIM protein 1		0.625	-
		MMP9	matrix metallopeptidase 9		−5.142	-
	LXR/RXR Activation	ARG2	arginase 2	−0.632	1.951	Activation
		MMP9	matrix metallopeptidase 9		−5.142	inhibition
		S100A8	S100 calcium binding protein A8		−3.185	inhibition
		SERPINF1	serpin family F member 1		0.963	Activation
	Cardiac Hypertrophy Signaling	ADRB2	adrenoceptor beta 2	1.265	0.800	Activation
		CACNA1S	calcium voltage-gated channel subunit alpha1 S		−3.299	Activation
		GNB3	G protein subunit beta 3		0.991	Activation

## Data Availability

The study data are available from the corresponding author upon reasonable request. The data discussed in this manuscript have been deposited in NCBI’s Gene Expression Omnibus (GEO) and are accessible through GEO Series accession number PRJNA739538 (https://www.ncbi.nlm.nih.gov/bioproject/?term=PRJNA739538, accessed on 18 October 2021).

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
