# Peer review of "Sarcomeres Morphology and Z-Line Arrangement Disarray Induced by Ventricular Premature Contractions through the Rac2/Cofilin Pathway"

_ijms, 2021, doi:10.3390/ijms222011244_

Round 1

Reviewer 1 Report

This is a very interesting paper looking at the role of disarrayed sarcomere morphology and Rac2/cofilin pathway in a pig model of VPCs.

Apart from some minor inaccuracies in the writing and punctuation, outlined below, the study represents a carefully planned and performed research worthy of publication.

  • references in square brackets shall be moved before the pagsemicolon/point
  • 3, Figure 2B, to change LVFM with LVFW
  • 4, line 111, to change LVF with LVFM
  • 4 (lines 136,138,139) and pag. 11 (line 37), typeface needs to be aligned to test for underlined words
  • in legends of Figures 4, 5 and 7 the P values are missing
  • 11, test from line 37 to 49 needs to be formatted
  • 13, line 105, “alteration” instead of “ alternation”
  • 13, line 115, reference [8] does not seem to represent the latter information about the heart
  • 13, line 121, “by the increased” instead of “by increased the”
  • 5 paragraph typeface needs to be aligned to test
  • In legends of Figures with western it needs to add “PC:positive control”

Reviewer 2 Report

This is an important experimental study in animal models to determine the underlying pathology of PVCs.

I congratulate authors to conduct this experiment. There are few reservations on this study.

  1. Study conclusion can not be generalized as it is limited to 6 animals.
  2. The stimulus used is electrical current for 6 months which produced significant changes in the myocardium, which is not the case in isolated PVC's.
  3. The authors have studied the effect on actin myofilaments along the Z line, but not studied the beta receptors, whether there is any upregulation or down regulation of beta receptors
  4. The findings of RAc2 played a crucial role has not been confirmed in knock out model, which limits the study outcomes.
  5. Although Rac2?cofilin was found in this study to be important, other pathways or mechanisms are not studied.
